# Microbial Evaluation of Ozone Water Combined with Ultrasound Cleaning on Crayfish (*Procambarus clarkii*)

**DOI:** 10.3390/foods11152314

**Published:** 2022-08-03

**Authors:** Yuzhao Ling, Hongyuan Tan, Lingwei Shen, Lingyun Wei, Guangquan Xiong, Lan Wang, Wenjin Wu, Yu Qiao

**Affiliations:** 1Key Laboratory of Cold Chain Logistics Technology for Agro-Product, Ministry of Agriculture and Rural Affairs, Institute of Agricultural Products Processing and Nuclear Agricultural Technology, Hubei Academy of Agricultural Sciences, Wuhan 430064, China; lingyuzhao2012@163.com (Y.L.); tanhongyuan1206@163.com (H.T.); shenlingwei2021@163.com (L.S.); xiongguangquan@163.com (G.X.); lilywang_2016@163.com (L.W.); 272081603@163.com (W.W.); 2School of Environmental Ecology and Biological Engineering, Wuhan Institute of Technology, Wuhan 430205, China; lingyun.wei@outlook.com; 3School of Bioengineering and Food, Hubei University of Technology, Wuhan 430068, China

**Keywords:** crayfish, ozone water, ultrasound, 16S rRNA gene sequencing

## Abstract

The effects of ozone water (OW) and ultrasound cleaning (UL) on microbial community diversity of crayfish were studied through microbial viable count and 16S rRNA gene sequencing. The results showed that compared with the control (CK), the ozone water combined with ultrasound cleaning (OCU) showed a significant reduction (*p* < 0.05) in total viable count (TVC), psychrophilic viable count (PVC), mesophilic viable count (MVC), Pseudomonas, hydrogen sulfide-producing bacteria (HSPB), molds and yeasts. Concretely, the TVC of the CK, OW, UL and OCU were 5.09, 4.55, 4.32 and 4.06 log CFU/g, respectively. The dominant bacterium in untreated crayfish was *Chryseobacterium*, and its relative abundance was reduced by combined treatment. Color measurement and sensory evaluation suggested that a satisfactory sensory experience could be obtained on the crayfish applied with OCU. In brief, OCU could be used as a cleaning strategy to control the microbial quality of crayfish and have no influence on its quality.

## 1. Introduction

Crayfish (*Procambarus clarkii*) is an important part of the fishery trade, and it is related to the outbreak of foodborne diseases [1], as well as crayfish plague [2]. The food-borne pathogens associated with crayfish mainly include *Vibrio*, *Listeria*, *Salmonella*, and *Shigella*. In order to solve the hidden dangers of food-borne pathogenic bacteria, cleaning and disinfection are considered essential to control the microbiological quality of crayfish.

Considering live crayfish, non-thermal sterilization is the first choice. Common physical food processing technologies have cold plasma, irradiation, high pressure, ozonation, ultrasound, ultraviolet light, pulsed light, and pulsed electric fields [3]. Ultrasound and ozone are considered harmless, effective and economical methods, and are widely used in food technology. As an emerging green technique, ultrasound is considered as a safe and efficient tool in food processing industry [4,5,6]. The wide application of ultrasound treatment was attributed to the cavitation effect caused by ultrasonic waves propagating in liquid systems, as well as thermal effect, shear force, micro-jet, and shock wave [7,8,9]. Ultrasonic inactivation of microbes is mainly due to cell rupture, localized high temperature and pressures, and the generation of several free radicals [9,10,11,12,13]. Kordowska–Wiater et al. reported that sonication of chicken wings in water for 3 min resulted in a reduction of bacteria on the skin surface, and the most sensitive was *Escherichia coli* (*E. coli*) [14]. However, ultrasound alone cannot significantly reduce microbial contamination [3]. The study suggested that bacteria were mostly washed away rather than destroyed by ultrasound in water [14]. Hurdle technology is feasible, which is defined as cleaning of ultrasound combined with other detergents. As a highly effective strong oxidant [15], ozone could inactivate a variety of fungi, Gram-positive bacteria, Gram-negative bacteria, and viruses [16,17]. Ozone use in the food industry has been reported. Rodrigues et al. indicated that ozone water immersion treatment could remove some pesticides (azoxystrobin, chlorothalonil, and difenoconazole) remaining in tomatoes [18]. Disinfection of red-meat-processing wastewater using ozone showed that 99% of TVC, total coliforms, and *E. coli* were inactivated [19]. Ozone spray effectively reduced the initial load of aerobic bacteria in refrigerated salmon fillets, and did not significantly increase the level of lipid oxidation [20]. Therefore, ozone water could be used as a disinfectant and washing liquid to facilitate subsequent food processing [21]. Studies have shown that ultrasound combined with ozone could be used to increase oxidative capacity, thereby providing a faster degradation rate of organic pollutants [22,23].

In recent years, the use of 16S rRNA gene sequencing has become popular in the analysis of bacterial community diversity; it is a highly sensitive and relatively quantitative technique [24]. Cleaning is one of the pre-treatments in the processing of crayfish, and it is the basic step to ensure the safety and quality of crayfish. Therefore, studying the bacterial spectrum in crayfish after pre-cleaning is integral to quality control. According to our current knowledge, ozone combined with ultrasound has been expanded in various areas, including device descaling [25], water sterilization [26], fruit preservation [27], biosludge reduction [28], and degradation of antibiotic drugs [29], but rarely in the cleaning and disinfection of crayfish. In this study, the effectiveness of ozone water combined with ultrasound cleaning in crayfish via microbiological (traditional plate count and 16S rRNA gene sequencing) and physicochemical parameters (color test, electronic nose, and sensory evaluation) were studied.

## 2. Materials and Methods

### 2.1. Materials

Crayfish were purchased from the Baili Supermarket, Hongshan District, Wuhan, China. The mean length and weight of crayfish were 13.0 ± 0.5 cm and 13.2 ± 0.4 g, respectively. The purchased crayfish were immediately transported to the laboratory in no more than one hour. The obtained crayfish were not pretreated before cleaning.

### 2.2. Treatment of Crayfish

Live crayfish were divided into four treatments, with 30 in each. Ozonizer (GCQJ-1-3, Wuhan, China), air-liquid mixer (HPSJ-25, Wuhan, China) and ultrasonic bath (KQ-500VDV, Kunshan, China) were applied: the concentration of ozone water and the intensity of ultrasound were 26.60 mg/L and 200 W, respectively. The batches were respectively washed by different cleaning methods with treatment groups as follows: (a) OW, immersed in ozone water for 20 min, (b) UL, ultrasound cleaning with ultra-pure water for 10 min, and (c) OCU, at first immersed in ozone water for 20 min followed by ultrasound cleaning with ultra-pure water for 10 min. In addition, crayfish immersed in ultra-pure water for 20 min set as the control (CK). The whole crayfish is covered with the soaking solution. The temperature of the ultrasound cleaning solution was kept at about 25 °C by means of an ice bath. After that, treated crayfish were used for microbiological and physicochemical analysis.

### 2.3. Microbial Count

Cleaned crayfish head and shell were removed. Precisely 5.0 g of shredded crayfish tail muscle with 45 mL of sterilized 0.85% (*w*/*v*) NaCl solution was stirred for 2 min. Sterilized glass beads were added to thoroughly contact the mixture. The suspensions were serially diluted in tubes with 9 mL of sterilized NaCl solution, then 1 mL suitable dilution was poured into petri dishes with agar medium, making the suspension and medium mix well by rotating the petri dish slightly. Total viable count (TVC) was cultivated by plate count agar (Hopebio, Qingdao, China) at 30 °C for 72 h; similarly, psychrophilic viable count (PVC) and mesophilic viable count (MVC) were incubated at 4 °C for 7 d and at 37 °C for 48 h, respectively. *Pseudomonas* were cultivated with cetrimide fucidin cephaloridine agar (Hopebio, Qingdao, China) with selected medium additives at 30 °C for 72 h. Hydrogen sulfide-producing bacteria (HSPB) were cultivated with triple sugar iron agar (Hopebio, Qingdao, China) at 30 °C for 72 h (black colonies formed due to the precipitation of iron sulfide). Molds and yeasts were incubated on rose bengal medium (Hopebio, Qingdao, China) at 28 °C for 5 d. Each crayfish sample was used for determination in duplicate. The number of cultured colonies was converted to colony-forming units per gram of sample (CFU/g).

### 2.4. DNA Extraction and Amplification

Twenty-four swabs were prepared containing surface inclusions of crayfish muscle from four treatments. E.Z.N.A @ Soil DNA kit (Omega Bio-Tek, Norcross, GA, USA) was used to extract the total DNA of crayfish samples, followed by using 1.2% agarose gel electrophoresis to determine DNA quality. The DNA concentration was measured by using BioSpec-nano (Shimadzu, Japan). The PCR mix is as described above, containing PCR ExTaq Buffer, DNA template, dNTP, ExTaq, primer1, and primer2. Universal bacterial primers are 338-F (5’-ACTCCTACGGGAGGCAGCA-3’) and 806-R (5’-GGACTACHVGGGTWTCTAAT-3’), which are used to amplify the V3-V4 region of the bacterial 16S rRNA gene. The PCR amplified conditions were: 95 °C for 2 min, and followed by 30 cycles: degeneration at 95 °C for 1 min, renaturation at 60 °C for 40 s, and elongation at 70 °C for 40 s, and finally extension at 70 °C for 10 min. The PCR products were identified by 1.2% agarose gel electrophoresis, and purified through the AxyPrep DNA Gel Extraction Kit (Axygen Biosciences, Union City, CA, USA). The PCR-Free Sample Preparation Kit (Illumina, San Diego, CA, USA) was used for the generation of amplicon library, and the Illumina HiSeq platform (Beijing Novogene Bioinformation Science and Technology Co. Ltd., China) was used for high throughput sequencing. This information was collected on the online platform from Shanghai Majorbio Bio-pharm Technology Co. Ltd. (Shanghai, China)

High-quality sequences were stitched by Flash software (version 1.2.11, https://ccb.jhu.edu/software/FLASH/index.shtml accessed on 20 June 2022), using a similarity level of 97%, and using Uparse software (version 7.0.1090, http://www.drive5.com/uparse/ accessed on 20 June 2022) to classify all sequences into operational tax units (OTU). OTU can be a genus or a phylum. In order to obtain the species classification information corresponding to each OTU, the silva132/16S_bacteria database was used for taxonomic comparison of species classification, together with the Ribosomal Database Project (RDP) classifier (version 2.11, https://sourceforge.net/projects/rdp-classifier/ accessed on 20 June 2022). This process was carried out on the platform for quantitative insights into microbial ecology (Qiime) (http://qiime.org/scripts/assign_taxonomy.html, accessed on 6 November 2019). The confidence of species classification was set as 70%. Mothur software (version 1.30.2, https://www.mothur.org/wiki/Download_mothur accessed on 20 June 2022) was used to evaluate alpha diversity of each sample, namely Coverage, Shannon index, Simpson index, Ace index, and Chao index, and the significance level was set by the Duncan method. Mapping with the online platform from Shanghai Majorbio Bio-pharm Technology Co. Ltd. (Shanghai, China), included Venn plot, bar-plot, heat-map, and principal co-ordinates analysis (PCoA) at bacteria community level.

### 2.5. Examination of Chromaticity

The L* (brightness value), a* (red or green value), and b* (yellow or blue value) of crayfish were measured by a portable colorimeter (CR-400, Konica Minolta Inc., Japan), and the colorimeter was corrected by a white standard board (L* = 85.6, a* = 0.3162, b* = 0.3238) before testing. Whiteness was used to distinguish the color difference between crayfish samples. The muscle chromaticity of four crayfish was measured, and the muscle test parts of each crayfish were randomly selected.
$$\mathrm{Whiteness}=100-\sqrt{{\left(100-L\right)}^{2}+a^{2}+b^{2}}$$

### 2.6. E-Nose Analysis

The PEN3 electronic nose (AIRSENSE, Schwerin, Germany) was used to analyze the odor difference of crayfish samples. 3 g of shredded crayfish muscle and 6 mL of saturated saline were put into the 20 mL headspace bottle, and then sealed with an E-Z Crimper (Huifen CO. Ltd., Zaozhuang, China). The headspace bottle was placed in a 45 °C water bath to equilibrate for 3 min, and then analyzed in the E-nose system. The test parameters were as follows: flush time and measurement time were set as 100 and 120 s, respectively, and chamber flow was set as 600 mL/min. The E-nose system consists of 10 metal oxide sensors, which are sensitive to different volatile components. As a result of previous experiments, the response values of the sensor from 90 to 94 s were used to visualize the odor difference between the samples, and this process was completed in the software WinMuster (Version 1.6.2, Schwerin, Germany) built into the E-nose system. Each sample was tested in triplicate.

### 2.7. Sensory Evaluation

Sensory evaluation was determined from three indexes: smell, color and muscle tissue, which were slightly modified [30]. Each indicator included three summary descriptive words based on sensory responses. The descriptors respectively represented three quality levels, which were excellent (representing 8 to 9 scores), good (representing 6 to 7 scores), and general (representing 5 to 6 scores). Seven experienced food professionals (4 males and 3 females) aged about 24 in the laboratory scored the descriptors, and the results were averaged.

### 2.8. Statistical Analysis

The results were expressed as means ± standard deviations derived from replicates. Origin 9.4 (Origin Lab, Northampton, MA, USA), Microsoft Excel 2010 and SPSS 18.0 (SPSS Inc., Chicago, IL, USA) were used for data analysis. One-way analysis of variance was carried out by Tukey multiple comparison method to evaluate the significance of differences between the data, and the value of *p* < 0.05 was considered as significant.

## 3. Results

### 3.1. Microbial Count

The MVC of OW was equivalent to that of UL (4.77 and 4.69 log CFU/g, respectively), but it decreased significantly (*p* < 0.05) in the combined treatment (4.24 log CFU/g). TVC (4.32 log CFU/g) was less than MVC (4.69 log CFU/g) in UL, which may be because the local high temperature generated by ultrasonic cavitation was not conducive to TVC growth. Compared with the control, using three cleaning methods (OW, UL, and OCU) significantly reduced (*p* < 0.05) the amount of TVC, MVC, PVC, *Pseudomonas*, HSPB, molds and yeasts (Figure 1). Most of the selective count in OCU showed a synergistic decline (*p* < 0.05), such as TVC, MVC, and PVC. Similarly, reduction of total coliform count can be observed in single treatments (ozone or ultrasound), but the combination showed a synergistic effect [31]. These counts were in agreement with previous studies. Compared with the control, ozone water reduced the TVC of tilapia fillets by 79.49% [32]. The microbial load of crayfish treated with ozone was significantly reduced [33,34]. Similarly, on day 0, the MVC of samples treated with ozone water was less than 1.40 log CFU/g, which was significantly lower than that of the control (2.97 log CFU/g) [35]. Ozone water spraying significantly reduced TVC of salmon fillets [20]. In contrast, the growth of psychrophilic bacteria was not observed in shrimp treated with ozone water on day 0 [35].

For both *Pseudomonas* and HSPB, the trends were similar. *Pseudomonas* and HSPB in OCU (3.53 and 4.02 log CFU/g, respectively) were significantly (*p* < 0.05) lower than levels in the control (4.03 and 4.50 log CFU/g, respectively). Similar treatment showed that *Pseudomonas* did not show resistance to the combined treatment of ultrasound with lactic acid solution [14]. On the whole, the level of molds and yeasts was lower by 2 to 3 orders of magnitude than that of bacteria. However, compared with the control, UL did not significantly reduce (*p* > 0.05) the amount of molds and yeasts, while a significant reduction occurred in OCU, which showed that the ability of ozone water to inactivate molds and yeasts was better than that of ultrasound. Both gaseous and aqueous ozone reduced the amount of yeast by about 0.5 log CFU/mL [36]. Some non-thermal technologies, such as high pressure and pulsed electric fields cannot effectively kill spores [3]. It was found, however, that ozone treatment could completely eliminate yeast and bacteria [37]. For the reduction of yeast, the effect of ozone was not affected by its form or action time. There was no significant difference in total yeast count between use of ozonated water and gaseous ozone, and the count was also not affected by the extension of reaction time [36]. Previous studies suggested that ultrasound has no direct effect on spores or Gram-positive bacteria [12], and that Gram-negative bacteria are more sensitive to ozone than Gram-positive bacteria [38], and finally yeast [39]. In addition, vacuum packaging after ozonation seems to be effective in reducing yeasts and molds [40], which may be due to the inability of most aerobic yeasts and molds to grow normally.

### 3.2. Bacterial Diversity

#### 3.2.1. Alpha Diversity

Illumina MiSeq high-throughput sequencing was used to evaluate the bacterial community diversity in crayfish samples. 1224285 effective 16S rRNA gene sequences (the reads from CK, OW, UL, and OCU were 293,960, 327,539, 302,607, and 300,179, respectively) were obtained from 24 crayfish samples in sextuplicate. Alpha diversity analysis was used to evaluate richness and diversity in the dominant microbiota of crayfish samples in this study, including Shannon index, Simpson index, Ace index, and Chao index. The average coverage from crayfish samples was higher 99% (data not shown), which suggested that the sequence reads almost characterized the microbial community of crayfish samples. As shown in Table 1, compared with CK, the Ace index and Chao index of both OW and OCU were significantly lower (*p* < 0.05), while UL decreased insignificantly (*p* > 0.05), which indicated that ozone water causes a more effective reduction of microbiota richness than ultrasound. In addition, the Simpson index of OCU (0.08) was the highest among the treatments, which illustrated that ozone water combined with ultrasound cleaning is the most beneficial for reducing the microbial community of crayfish.

#### 3.2.2. Bacterial Community Composition

Most OTU in crayfish belonged to five bacteria phyla, including *Proteobacteria*, *Bacteroidetes*, *Deinococcus-Thermus*, *Actinobacteria*, and *Firmicutes*. *Bacteroidetes*, *Actinobacteria*, and *Firmicutes* were reduced to varying degrees by ozone water, ultrasound, and the union of both (Figure 2A). Compared with the control, the *Proteobacteria* from three treatments increased markedly. *Deinococcus*-*Thermus* from both OW and OCU was also significantly decreased (4.23 and 4.27%, respectively) compared with the control (5.05%), while UL (5.90%) was increased. These phyla include *Proteobacteria*, *Bacteroidetes*, *Actinobacteria*, and *Firmicutes*, which usually appear alone or in combination in the microbiota of crayfish and shrimp [41,42,43,44,45]. In fact, *Proteobacteria*, *Firmicutes*, *Actinobacteria*, and *Bacteroidetes* were also dominant phyla in other food samples and meat processing rooms [37,38,46].

As shown in Figure 2B, twenty genera of total relative abundance value > 1% were identified at the genus level. The relative abundance of *Cloacibacterium*, *Soonwooa*, *Leucobacter*, and *Comamonas* was decreased after ultrasound cleaning alone; in detail, the values were 4.58, < 1, 0.68, and 2.94%, respectively, lower than that of the control (5.51, 1.56, 0.72, and 5.11%, respectively). As previously mentioned, the sterilization mechanism of ultrasound was mainly generated by free radicals that damage cell membranes and heat-inactivate enzymes [11]. However, *Escherichia*-*Shigella* was an exception, and its sequence reads increased rapidly in UL, while OTU level in CK was lower. Studies reported that *Escherichia-Shigella*, which belongs to the opportunistic pathogens [47], was significantly enriched after ultrasound treatment [48,49]. *Escherichia-Shigella* has a tenacious ability to survive, and its relative abundance was increased after the action of disinfectants such as ozone, acid solution, chlorine, and antibiotics [50,51,52].

The relative abundance of genera in OW decreased from *Chryseobacterium*, *Deinococcus*, *Sphingobacterium*, *Hydrogenophaga*, *Taibaiella*, *Leadbetterella*, and *Acinetobacter*. Furthermore, these genera showed synergistic effects in OCU. *Chryseobacterium* derived from the phylum *Bacteroidetes* was the second largest dominant genus in abundance level. Its abundance in CK, OW, UL, and OCU were 17.42, 14.49, 13.59, and 11.93%, respectively. This result indicated that the best reduction effect occurred in ozone water combined with ultrasound, followed by ozone water, and finally ultrasound. Similarly, cleaning treatments were performed on strawberries containing artificially inoculated *E. coli*, and it was found that ultrasound treatment could not completely remove the inoculum, but ozone could, and the combination of the two showed a synergistic reduction effect [27]. For the osmotic lysis of bacteria, although ultrasound has a mechanical effect, O_3_ has a chemical effect that causes a significant increase in the production of reactive oxygen species [53]. Hydroxyl radicals have a powerful oxidizing capacity, can instantaneously penetrate into the cell, and then act on the components of cytoplasmic membrane and intracellular enzymes (such as superoxide dismutase and catalase) [54]. Likewise, Walker et al. used a presoak solution before ultrasound cleaning, producing the most effective instrument cleaning effect on the whole, compared to presoak cleaning only and ultrasound cleaning only [55]. For the two cleaning treatments (ultrasound and plasma liquids), synchronous treatment reduced the amount of *E. coli* and *Shewanella putrefaciens* more than that of separate treatment [56]. Ozone water combined with ultrasound enhanced the antibacterial ability of ozone and resulted in >3 log reduction of microorganisms in cherry tomato [57]. Due to the local high temperature and high pressure generated by ultrasonic cavitation, the bacterial cell membrane becomes fragile and hydroxyl radicals generated by ozone decomposition can penetrate into the cell more easily [58]. In addition, studies showed that ozone could react with biofilm components [59] and ultrasound could enhance the capability of ozone to reduce the thickness of biofilm [25].

Compared with CK (8.96%), the relative abundance of *Pseudomonas* in treatments increased. Concretely, the abundance in OW, UL, and OCU were 14.83, 19.41, and 23.22%, respectively. Generally, *Pseudomonas* was considered to be the dominant microbiota in crayfish [45,60]. However, because the total microbial load was unknown, although the relative abundance of *Pseudomonas* increased, it was not known whether this phenomenon was due to an increase in its absolute quantity or a decrease in the abundance of other microbiota. This phenomenon has occurred in other species. *Staphylococcus* counted by microbial plate was not affected by ozone treatment, but its relative abundance was decreased by high-throughput sequencing [38]. Taking *Geobacter* as an example, its detected level was extremely low (<1%) after ozone water or ultrasound cleaning compared with the control (2.59%), which indirectly increased the relative abundance of *Pseudomonas*. Another fact referred to the “other” category: the relative abundance of CK, OW, UL, and OCU were 24.03, 17.14, 20.23, and 15.93%, respectively, which indicated that some less abundant and unclassified genera in the “others” category were more sensitive to ozone water and ultrasound treatment.

#### 3.2.3. Sample Difference

There were significant defferences in OTU levels in the microbiota of crayfish treated with ultrasound, ozone water, and a combination of the two compared to the control (Figure 3A). In a Venn plot (Figure 3B), the microbial species of CK, OW, UL, and OCU were 1917, 1658, 2072, and 1684, respectively. The shared species in two comparisons, OW&OCU and UL&OCU were 37 and 108, respectively. Furthermore, the unique species in OW and UL were 159 and 309, respectively, which showed that reduced species occurred more in the ozone water than ultrasound. Consistent with the preceding results, the sterilization effect of ozone water is better than that of ultrasound. The principal co-ordinates analysis (PCoA) reflected the differences in microbial community between different cleaning methods (Figure 3C). The first principal component (PC1) and the second principal component (PC2) were 31.03% and 19.04%, respectively, which represent the dominant microbial species in crayfish samples. Cluster analysis showed that the species distribution between CK and treatments showed great isolation, which indicated that ozone water and ultrasound treatment could significantly reduce the microbial community of crayfish. The species distribution between OW and OCU did not show great isolation, which suggested that the inactivation capacity of the combined cleaning mainly owed to ozone water. There was a potential synergistic effect on bacteria reduction between ozone water and ultrasound, and further research is urgently needed to understand the detailed mechanism of the combined treatment.

### 3.3. Chromaticity

The color of food is an important parameter of consumer acceptance. As shown in Figure 4A, compared to the control, the a* values (redness) of OW and OCU decreased significantly (*p* < 0.05), which may be attributed to excessive protein oxidation caused by ozone water. UL contributed the largest a* value (4.55), which indicated that the ruptured cells formed by ultrasound released more haemoglobin and myoglobin pigments. The increased a* value of tilapia fillets was attributed to the mechanical effect of ultrasound to induce cytochrome release [61]. Although OCU reduced redness, its yellowness is the lowest among all groups (Figure 4B). Generally speaking, low yellowness is a desirable result. Ozone water bleached the crayfish, but ultrasound treatment made up for this defect by increasing redness. Therefore, the lightly increased whiteness (43.96) caused by combined treatment was not significant (*p* > 0.05) compared with the control (43.30) (Figure 4C). Similarly, Esua et al. found that plasma functionalized liquid combined with ultrasound treatment of grass carp led to an increase in whiteness. In brief, crayfish cleaned by ozone water combined with ultrasound have obtained an acceptable appearance [56].

### 3.4. E-Nose

Electronic nose technology is considered as a simple and fast parameter for characterizing complex odors in food samples [62]. Figure 5 shows the odor clustering of crayfish samples after different cleaning methods. The sample variances of PC1 and PC2 were 75.30% and 20.40%, respectively, which almost characterized the volatile odor components of all crayfish samples. The relatively larger isolation areas showed that the odor components of three treatments (OW, UL, and OCU) were significantly different from the control. The overlapping areas were shared between the ozone water alone, the ultrasound alone, and their combination. Wenzheng et al. reported that the content of odor compounds (for example geosmin), which contributed a lot to the fishy smell in aquatic products, was significantly reduced after ozonation [63]. As mentioned earlier, the cavitation effect of ultrasound could synergistically degrade and release contaminants attached to the surface of crayfish [8]. Ozone water combined with ultrasound cleaning changed the distribution of odor compounds in crayfish, so it is necessary to carry out research on the effect of combined treatment on fishy components.

### 3.5. Sensory Evaluation

As shown in Figure 6, UL has the highest color score (8.14), which may be due to ultrasound destroying muscle cells and freeing myoglobin. The light red color (6.14) of OW was attributed to the strong oxidation of ozone water. In terms of texture score, there was no significant difference (*p* > 0.05) between treatments. Concretely, CK, OW, UL, and OCU scored 7.43, 6.57, 6.57, and 6.71, respectively. The intrinsic unpleasant odor of ozone water resulted in a lower odor score (6.71) in OW, while a higher odor score (7.57) in OCU may be attributed to the cavitation effect of ultrasound. In a word, crayfish cleaned by ozone water combined with ultrasound have obtained satisfactory sensory experiences, including color, texture, and smell.

## 4. Conclusions

Ozone water combined with ultrasound cleaning reduced the load of TVC, PVC, MVC, *Pseudomonas*, HSPB, molds and yeasts in crayfish. Color test, E-nose analysis, and sensory evaluation also showed that the combined treatment would not adversely affect the quality of crayfish. In brief, the combined cleaning can be used as a pretreatment method for crayfish products to reduce bacterial contamination and improve the quality of crayfish. In addition, there was a potential synergistic effect between ozone water and ultrasound in bacteria reduction. Thus, further research is urgently needed to understand the detailed mechanism of the combined treatment.

## Figures and Tables

**Figure 1 foods-11-02314-f001:**
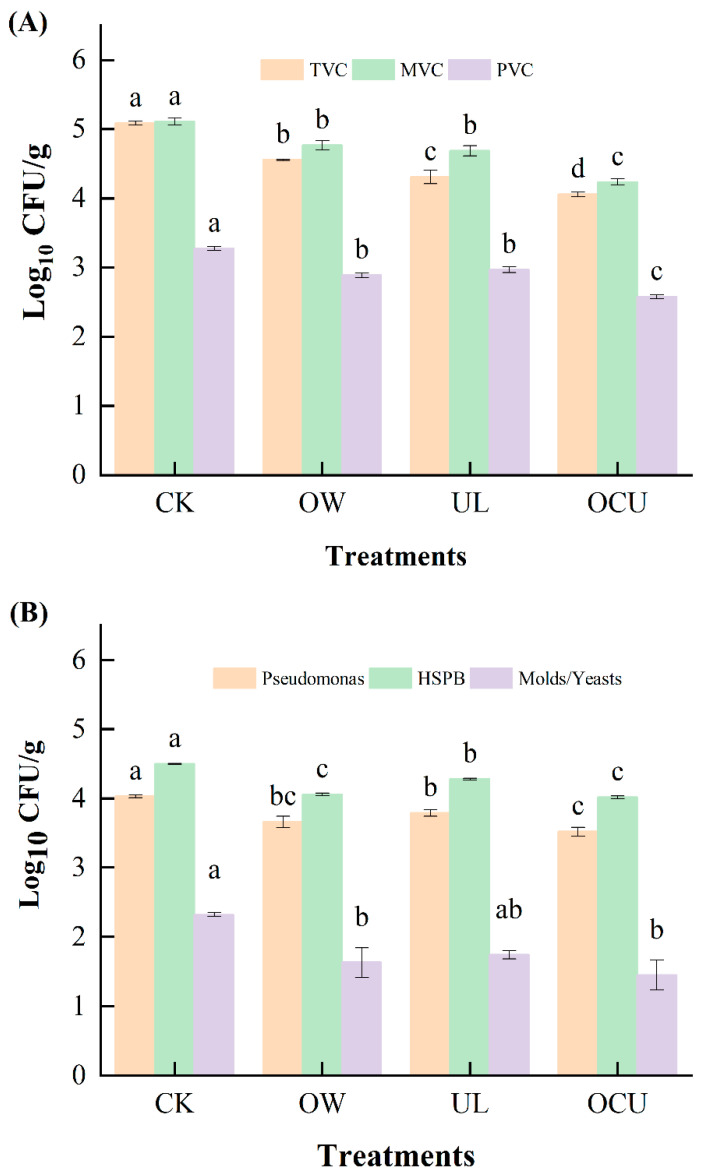
Panels (**A**,**B**) show the microbial count of crayfish washed by different treatments; mean values are expressed as log CFU/g. Error bars were derived from standard deviation of means. The significant differences (*p* < 0.05) between treatments for the same index are expressed alphabetically (a, b, c, and d). Notes: CK, the control; OW, ozone water cleaning; UL, ultrasound cleaning; OCU, ozone water combined with ultrasound cleaning; TVC: total viable count; MVC: mesophilic viable count; PVC: psychrophilic viable count; HSPB, hydrogen sulfide-producing bacteria.

**Figure 2 foods-11-02314-f002:**
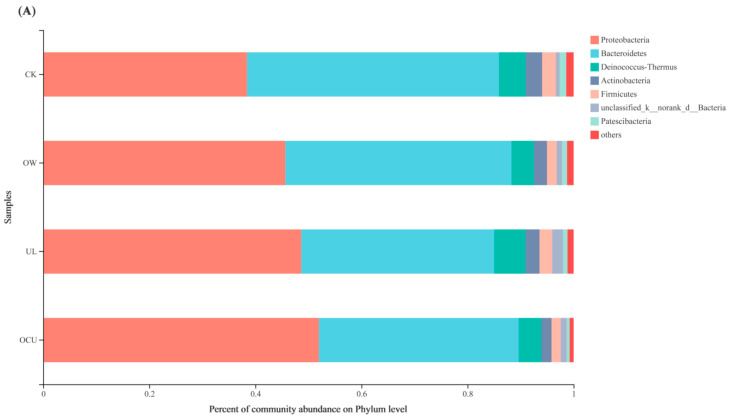
The relative abundance of crayfish washed by different treatments at the phylum (**A**) and genus (**B**) level. All genera with relative abundance less than 1% were classified as “other”. Notes: CK, the control; OW, ozone water cleaning; UL, ultrasound cleaning; OCU, ozone water combined with ultrasound cleaning.

**Figure 3 foods-11-02314-f003:**
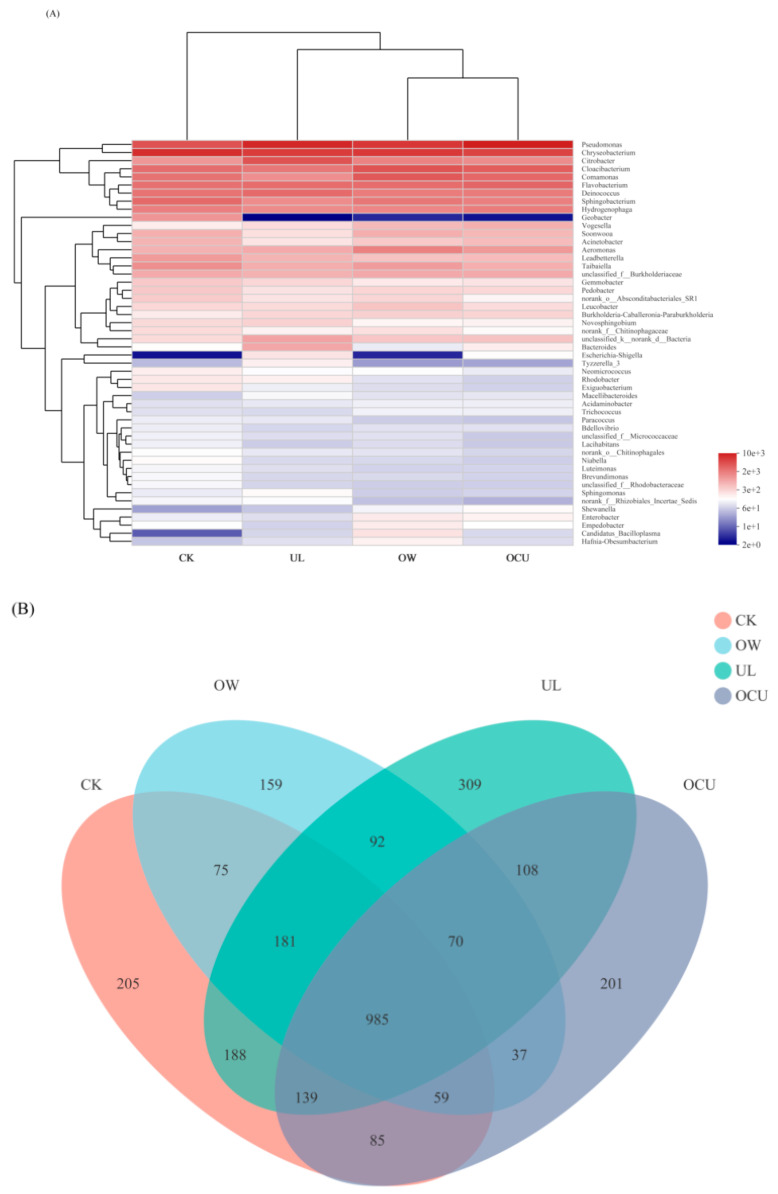
Panels (**A**–**C**) were respectively the heat-map, Venn plot, and principal co-ordinates analysis (PCoA) of crayfish (readers can choose the online version of this article to refer to the colors in the heat-map). Notes: CK, the control; OW, ozone water cleaning; UL, ultrasound cleaning; OCU, ozone water combined with ultrasound cleaning.

**Figure 4 foods-11-02314-f004:**
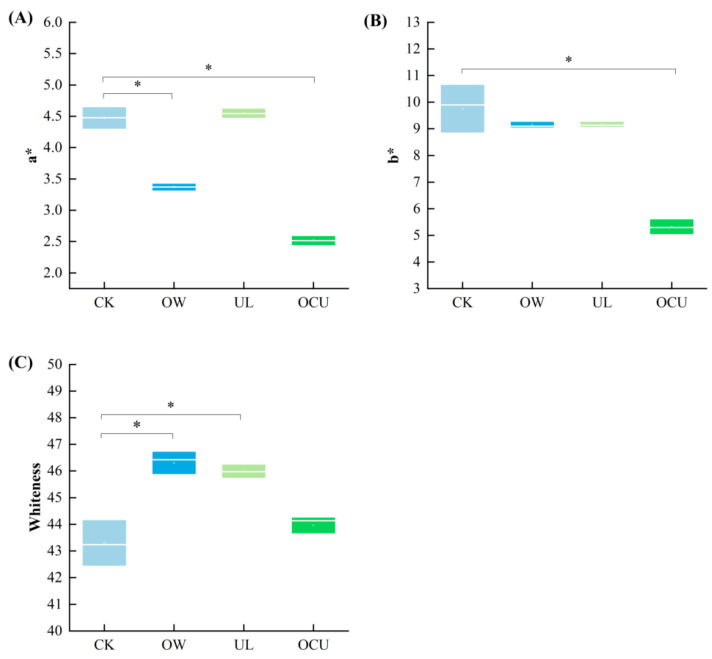
Panels (**A**–**C**) were respectively a* value, b* value, and whiteness, and expressed as boxplots. Compared with the control, the significant difference was expressed as * (*p* < 0.05). Notes: CK, the control; OW, ozone water cleaning; UL, ultrasound cleaning; OCU, ozone water combined with ultrasound cleaning.

**Figure 5 foods-11-02314-f005:**
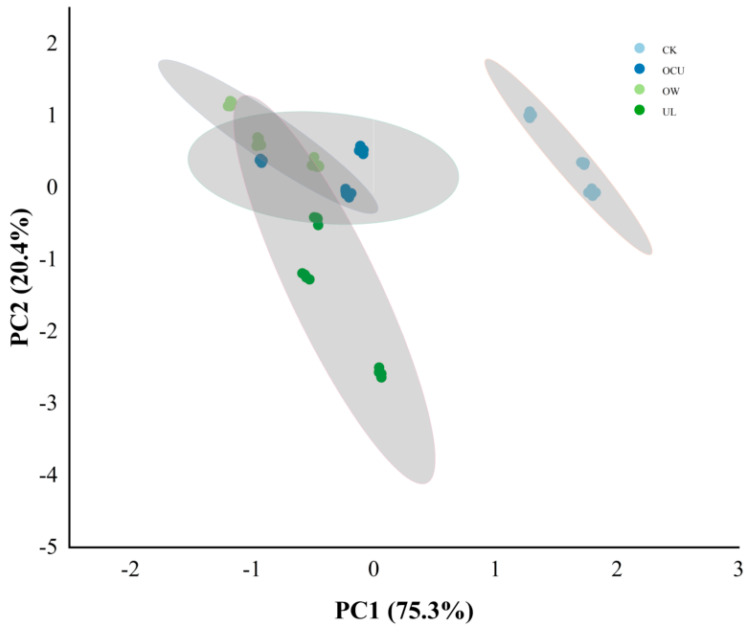
Crayfish odor clustering based on principal component analysis. Notes: CK, the control; OW, ozone water cleaning; UL, ultrasound cleaning; OCU, ozone water combined with ultrasound cleaning.

**Figure 6 foods-11-02314-f006:**
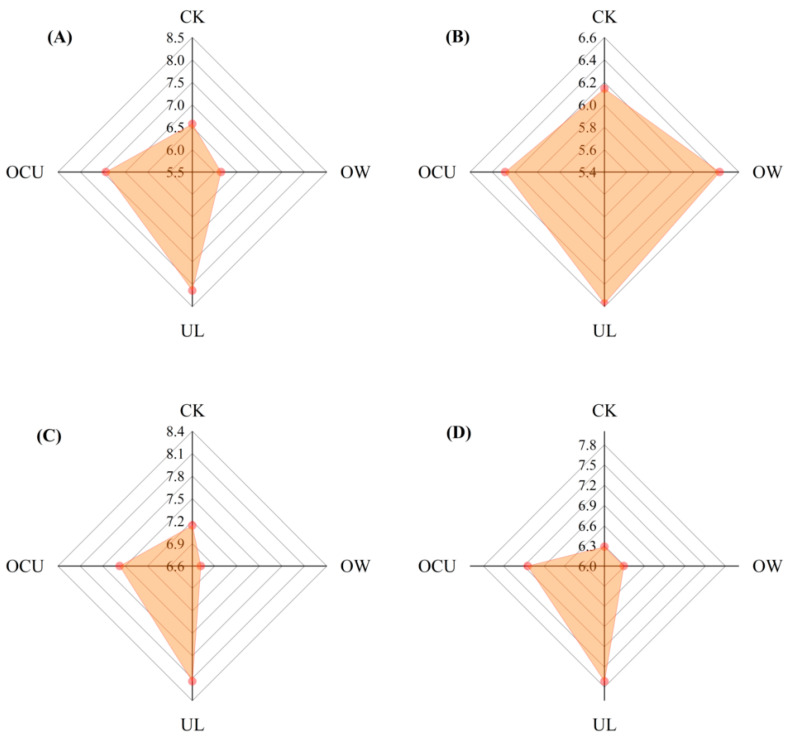
Panels (**A**–**D**) were respectively color, texture, odor, and overall evaluation of crayfish samples, and expressed by radar map. Notes: CK, the control; OW, ozone water cleaning; UL, ultrasound cleaning; OCU, ozone water combined with ultrasound cleaning.

**Table 1 foods-11-02314-t001:** The alpha diversity indexes of crayfish samples.

Samples	Shannon	Simpson	Ace	Chao
CK	4.46 ± 0.22 ^a^	0.03 ± 0.01 ^a^	1230.25 ± 133.13 ^a^	1235.84 ± 141.03 ^a^
OW	4.09 ± 0.10 ^ab^	0.05 ± 0.01 ^ab^	996.00 ± 126.95 ^bc^	980.03 ± 121.72 ^bc^
UL	4.20 ± 0.34 ^ab^	0.06 ± 0.03 ^ab^	1190.31 ± 128.74 ^ab^	1173.43 ± 123.24 ^ab^
OCU	3.91 ± 0.26 ^b^	0.08 ± 0.03 ^b^	943.25 ± 91.79 ^c^	930.18 ± 90.89 ^c^

Notes: The significant difference (*p* < 0.05) between treatments for the same index was expressed alphabetically (a, b, and c). CK, the control; OW, ozone water cleaning; UL, ultrasound cleaning; OCU, ozone water combined with ultrasound cleaning.

## Data Availability

The datasets generated during and/or analyzed during the current study are available from the corresponding author on reasonable request.

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
