# Peer review of "Microbial Evaluation of Ozone Water Combined with Ultrasound Cleaning on Crayfish (Procambarus clarkii)"

_foods, 2022, doi:10.3390/foods11152314_

Round 1

Reviewer 1 Report

The manuscript is well written and the working plan well executed. However, some improvements are needed. 

Materials and methods

More information regarding the ultrasound technology ie needed to make the data reproductable

Inform the kind of US equipment used (tip or bath)

The vessel volume and dimensions used as well as the potency density applied (W/L or W/mL)

Was the heat generated by US controlled or measured?

 Results

The microbial log reduction cannot be associated with sterilization (line 329) as mentioned in some parts of the text.  The log reduction even with the combined treatment was about 2 log cycles, and was strongly dependent on the microbial strain. Some considerations regarding the acceptable counting must be added to the manuscript regarding the shelf life extension of the product. Some considerations regarding the product shelf life after the treatment should be also done in the manuscript text.

Line: 295 – inform the temperature reached by sonication 

Figure resolution must be improved, most legends are illegible and the figures definition are bad.

Author Response

We appreciate your professional comments on our article. As you are concerned, there are several problems that need to be addressed. According to your nice suggestions, we have made extensive corrections to our previous draft and also attached a point-by-point letter to you.

Due to word compatibility reasons, our high-resolution images were not displayed. But we have provided a clear PDF version with pictures for you or readers to understand this work. In addition, we provide a zip file containing all the images into the system, so that the article can use high-resolution images after receiving them.

Reviewer 2 Report

The authors evaluated the effect of ozone water combined with ultrasound cleaning in reducing microbial population in crayfish. The authors have used conventional as well 16S sequencing to evaluate the effect. However, they are unable to relate their research design and results with food safety aspect of crayfish processing. 

Following are the specific comments: 

L39-80: Introduction: Please include food safety problems/outbreaks and human pathogens associated with crayfish. Also explain how the treatment results based on general microbial analysis can be linked to the pathogens reduction. Explain also the significance of including 16 S rRNA sequencing in this study. 

L-70: Which fungal pathogen could be a human illness concern?

L-92-93:  Only immerged in OW or shaking was also applied? 

L100-114: Did you measure microbial population before and after treatment? 

L100-10: Why was the sample size so low? 25 g is the standard sampling size. Was there any reason taking so low weight? Please explain a little about samples processing, especially stirring the samples in solutions. 

L-180: Should be Result and Discussion 

Fig1;  Is the statistical difference result between the treatments or within a treatment? There is not much difference, less than 0.5 log, if we compare the treatments with the control. How do you justify that the treatments were promising in reducing microbial load? 

What was the initial microbial count before applying treatments including Ultra-Pure water only ( CK)? You should have used different crayfish for different treatment,  and there is a probability that microbial population and community structure is different between the crayfish samples. How do  you justify generalizing the results considering all the tested samples had the same microbial population before treatments? 

L-184-186: What was the  temperature? Does this increased temperature  reduce the shelf-life of the product? 

L-239-241: There was no significant difference between OW , OCU and CK based on  Shannon index.  How did you conclude the results based on ACE and Chao index ignoring Shannon index? 

Methodology: did you measure the microbial population in wash solution aftertreatment? 

Please revise the results and discussion section explaining the results in food safety and quality ( may be shelf life) aspects. 

Author Response

(The authors gave the same response as above.)

Reviewer 3 Report

Thank you for the opportunity to review this manuscript. This study addresses an interesting topic by combining the effects of ozone water (OW) and ultrasound cleaning (UL) on the reduction of microbial community in crayfish, showing a potential synergistic effect between ozone water and ultrasound in bacteria reduction.

However the manuscript is badly written and the research results are scarcely presented. It is not clear the message the Authors want to address and the huge amount of data and graphics seems to have more the purpose of confusing the reader rather than guiding him/her towards understanding the work.

 Moever I was wondering if they  tried to understand whether the use of ultrasound could influence the properties of the ozonized water .

Finally they describe ozono as one of the “Common physical food processing technologies”, I would better classify ozone as a “chemical” substance.

Author Response

(The authors gave the same response as above.)

Reviewer 4 Report

This research paper deals with microbial evaluation of ozone water combined with  ultrasound cleaning on crayfish (Procambarus  clarkii). Ozone could inactivate fungi, gram-positive bacteria, gram-negative bacteria, and viruses. Ultrasound cleaning on crayfish via the combination of traditional plate count and 16S  rRNA gene sequencing, together with  color test, electronic nose analysis, and sensory evaluation were carried out to assess the quality of crayfish. In this respect, the originality of this paper is fair enough. Although there are some considerable concerns/amendments that need to be addressed. 

 Specific comments

 1.         Science is written in the third person so it must be not used "we" in the manuscript. Thus, please remove them from the entire the text of MS.

2.         The objectives of the paper are not well documented consistently to represent the whole idea so please reconstruct the objectives of the paper at the end of introduction section again.

3.         Although the manuscript was generally understandable, there are some grammatical and syntax errors in the text. Thus, the language of paper has to be checked carefully again.

4.         The abstract of the MS is not well documented to represent the whole manuscript objectives. Please revise it and add some important values found for the quality parameters.

5.         Abbreviations used in whole manuscript have to be defined firstly and then their abbreviations have to be used.

6.         Regarding to the originality of the manuscript, it should be indicated clearly how this manuscript contributes to the existing knowledge. How it will likely to contribute to the state-of-the-art?

7.         Conclusion is a just repetition so it is not suitable. What is the core message to give food industry?

8.         Please check the references again for inconsistencies according to the format of the journal.

Author Response

(The authors gave the same response as above.)

Round 2

Reviewer 2 Report

The Authors have addressed the comments. 

Reviewer 3 Report

Dear Authors,

following the suggestions of the referees you have sufficiently adjusted the paper.

Now it is suitable for publication

Reviewer 4 Report

Authors took all comments into consideration and improved scientific value of the paper.